# Vascular Complications Caused by Tibial Osteochondroma: Focus on the Literature and Presentation of a Popliteal Artery Thrombosis with Acute Lower Limb Ischemia

**DOI:** 10.3390/diagnostics12051191

**Published:** 2022-05-10

**Authors:** Andrea Angelini, Mariachiara Cerchiaro, Carlo Maturi, Pietro Ruggieri

**Affiliations:** 1Department of Orthopedics and Orthopedic Oncology, University of Padova, 35128 Padova, Italy; chiaracerchiaro@gmail.com (M.C.); pietro.ruggieri@unipd.it (P.R.); 2Department of Vascular, Endovascular Surgery, University of Padova, 35128 Padova, Italy; carlo.maturi@aopd.veneto.it

**Keywords:** osteochondroma, popliteal artery thrombosis, acute lower limb ischemia, treatment

## Abstract

Osteochondromas are common benign bone tumors, frequently found in adolescents or young adults. Most often asymptomatic and discovered by accidental findings, they may be diagnosed because of compression or dislocation. Vascular complications are an atypical presentation of osteochondromas, and include vessel perforation and thrombosis, arterial thromboembolic events and pseudoaneurysm formation. Popliteal artery thrombosis and acute lower limb ischemia caused by a tibial osteochondroma are rarely observed. Starting from a case of temporary lower extremity ischaemia caused by thrombosis of the subarticular popliteal artery due to an osteochondroma of the proximal tibial protruding in popliteal fossa, we focused a literature analysis on diagnostic and management aspects. A combined vascular-orthopedic approach was performed with intra-arterial locoregional thrombolytic therapy and then a surgical tangential resection of the tibial osteochondroma. The adequate approach for these patients includes clinical evaluation, plain radiographs, CT scan and MRI. The purpose of the present review article is to underline the importance of a combined vascular-orthopedic approach to correct diagnosis and prompt surgical management of vascular complications caused by tibial osteochondromas.

## 1. Introduction

Osteochondromas are common benign tumors of bone, most frequently found in adolescents or young adults. Osteochondromas can be sessile or pedunculated, presenting as a solitary lesion in 85% of the cases, while 15% occur in the context of hereditary multiple exostoses (HME), a genetic autosomal dominant disorder. Although often asymptomatic and discovered incidentally, symptoms may derive from compression or dislocation of near structures such as adjacent vessels or nerves, fractures, osseous deformities, bursa formation or malignant transformation [1]. Cartilage cap thickness > 2 cm in adults or >3 cm in children as well as new onset of pain or growth, or rapid growth of the lesion, especially after the closure of the growth plate, might reflect cancerous transformation [2]. The involvement of lower extremities is common, particularly metaphyseal structures of the femur and tibia around the knee joint [3,4]. Surgical resection is indicated for symptomatic lesions, complications, cosmetic reasons or malignant transformation. Local recurrence is less than 2% if complete resection is achieved.

Vascular complications are rare and include vessel perforation and thrombosis, arterial thromboembolic events and pseudoaneurysm formation [3]. From an epidemiologic point of view, pseudoaneurysms are most frequently observed compared to thromboembolic complications [5]. The pathogenesis and development of these vascular complications are unknown because the exostoses are usually not detected until the secondary symptoms appear. Only few cases with osteochondromas of the proximal tibial causing popliteal artery thrombosis and acute lower limb ischemia have been reported [6,7,8]. The purpose of the present review article is to underline the importance of a combined vascular-orthopedic approach to correct diagnosis and prompt surgical management of vascular complications caused by tibial osteochondromas.

## 2. Focus on the Literature and Search Strategy

A systematic search of the literature was done to identify studies reporting on patients treated for vascular complications secondary to the lower limb osteochondroma. English and non-English language literature were searched in Pubmed using the terms ‘osteochondroma [MeSH Terms]’, ‘vascular’, ‘complications’, ‘pseudoaneurysm’, ‘thromboembolic events’, ‘artery’, ‘lower limb’ and ‘lower extremities’ in different combinations and in ISI Web of Knowledge database. The search was done on literature published between 1965 and 2022, resulting in about 210 articles (mainly small series and case reports) describing over 60 cases (Table 1). Vascular complications in lower limb osteochondromas affected young people with a mean age of 20.9 years (range 9 to 51 years). Twenty-four patients (39%) had HME and 36 solitary exostosis (59%). There is a large predominance in men: 49 male patients (80%) and 12 female patients (20%). Symptoms include swelling and pain (27, 44%), swelling (10, 16%) or pain (9, 15%) only, Ischemia (10, 16%) and Claudication (2, 3.3%). Exostosis arises from femur (53), tibia (8) or fibula (1). Vascular complications consist of pseudoaneurysm, thrombosis or acute ischemia.

## 3. Representative Explanatory Case

A 34-year-old male was referred to the emergency department with acute left lower limb ischemia.

His clinical history showed a similar ischemia at the same lower left limb four years before and other temporary symptoms. Oncologic history included acute lymphoblastic leukemia treated with chemotherapy, radiotherapy and bone marrow transplantation, and thyroidectomy for papillary thyroid carcinoma. He reported a gradual decrease of walking distance with increasing pain, cold and functional impairment of lower limb. On clinical examination he presented ecchymosis at heel level (Figure 1a).

Physical examination revealed loss of popliteal and peripheral pulses whereas the ipsilateral femoral pulses were normal, with no neurological deficit of the affected limb. A CT scan revealed an osteochondroma on the posterior face of the tibial plateau, close to the vascular bundle (Figure 1b), and the CT angiogram showed thrombosis of the subarticular popliteal artery (Figure 1c). Plain radiographs (Figure 1d) and contrast-enhanced MRI showed the pedunculated tumor with a thin cartilaginous cup (Figure 2).

He was urgently treated by vascular surgeons with intra-arterial locoregional thrombolytic therapy in place for four days. At subsequent checks, occlusion remained in the upper third of the interosseous artery. Interestingly, at ultrasound doppler evaluation the blood flow disappears at complete extension of the knee and appears during flexion. The patient started oral anticoagulant therapy with enoxaparin 4000 UI twice a day at first, followed by warfarin for six months. Two months later, a surgical resection was performed in our orthopedic department: using a posteromedial approach, the bone surface was reached, the lesion was isolated and a tangential resection was performed (Figure 3a). Histo-pathology evaluation confirmed the diagnosis of osteochondroma with no malignant features. The postoperative course was uneventful with palpable distal pulses, decreasing oedema and marked pain relief. Physical rehabilitation was not required, with immediate full recovery resuming his regular activities. At two years follow up, the patient was asymptomatic and CT angiogram showed normal triphasic arterial waves in the peripheral arterial circulation (Figure 3b).

## 4. Discussion

Osteochondromas are the most common benign primary bone tumors in growing patients representing about 35–45% of all benign bone tumors. About 90% of these occur sporadically and are usually solitary [1,13,29,30,59]. Osteochondromas occur most often around the knee (40% of cases) and proximal tibial is affected in 15–20% of cases [59]. Symptoms are usually related with compression of close related structure: tumors can occasionally dislocate the adjacent vessels and cause various types of vascular injury such as pseudoaneurysms, arterial thrombosis or acute lower limb ischemia [4,6,13,29,30,56,59]. However, vascular complications are an atypical presentation of osteochondromas. The affected vessel is commonly the popliteal artery, which is involved in about 60% of the cases. Rarely, osteochondromas may be complicated by fracture secondary to strenuous physical exercise or trauma, resulting in sharp bone spicules that pierce the artery, causing pseudoaneurysm formation [30]. Sakata et al. reported a pseudoaneurysm of the popliteal artery in a 16-year-old boy that underwent emergency surgery. Authors concluded that all vascular surgeons should know this complication and perform early surgical intervention due to the risk of occlusion of peripheral branches, deep venous thrombosis and peripheral nerve disorders [60]. Shah at al. reported an unusual case of claudication in a young adult caused by multiple ostheocondromas [29]. Adult patients with asymptomatic osteochondromas may also present a pseudoaneurysm of the popliteal artery [29]. Pseudoaneurysms are the most common vascular complications caused by ostheocondroma, but thromboembolic complications are rare, such as lower limb ulcer [56]. The pathogenesis and development of these vascular complications are unknown because the exostoses are usually not detected until the secondary symptoms appear. Suggested mechanisms include ossification of the cartilage cap and trauma to the region either from blunt impact or vigorous exercise. Some authors reported a case of exercise-induced lower extremity claudication due to popliteal artery occlusion from a bony exostosis, in which symptoms began after the patient experienced an acute painful episode while running [61,62]. Deep venous thrombosis in association with osteochondroma is infrequent. The osteochondroma exerts mass effect on the vein inducing thrombosis [30]. In our patient, the symptoms are related to transient acute ischemia of the lower limbs and osteochondroma was occasionally found in differential diagnosis with other vascular diseases. Arterial pseudoaneurysms are the most common vascular complication observed in HME. The lesion mechanism is due to a repeated abrasion of the arterial wall by a prominent osteochondroma, which erodes the arterial surface and generates the pseudoaneurysm [58,63]. Even though these pseudoaneurysms can be presented in any artery of the body, the popliteal artery represents the most frequent affected vessel (77%) [58]. Shah at al. reported an unusual case of claudication in a young adult caused by multiple ostheocondromas [61].

## 5. Diagnosis

Pedunculated osteochondromas around the knee have a typical appearance, extending away from the adjacent joint, described as stalactites and stalagmites. In any bone that develops from enchondral ossification, osteochondromas have a marrow, cortical and periosteum continuity with the underlying native bone. This direct continuity clearly appreciable in CT scan is a pathognomonic finding [59]. The cartilaginous cup is the most important area that should be investigated. On MRI, the cartilaginous cap is typically hypointense on T1-weighted images and variable on T2 images (hypointense or hyperintense if densely or poorly mineralized, respectively) [59]. Angiography and other vascular studies should be considered in the presence of vascular compression or for surgical planning [7,12,19,39]. Sometimes these exams are helpful to characterize neovascularity, which may signal malignant transformation [7,19].

It is important to include bone tumors in differential diagnosis in patients with vascular symptoms. The adequate approach for these patients includes clinical evaluation, plain radiographs, CT scan and MRI. In the presence of bone tumors, CT and MRI are the most sensitive examinations to exclude malignant appearance, soft tissue extension and signs of popliteal entrapment syndrome (PES). In young adult patients presenting with intermittent claudication or vascular insufficiency mimicking atherosclerotic peripheral vascular disease, alternative causes such as trauma, tumors, thromboangioitis obliterans, fibromuscular dysplasia, vasculitis and PES should be suspected [56]. PES refers to a group of symptoms caused by mechanical compression of popliteal artery, vein or tibial nerve in the popliteal fossa. This is usually related to musculotendinous structures, but a few case reports described bony exostoses of the popliteal fossa causing PES [3,6]. Angiography and ultrasound are useful to define the precise level of vascular defects. In patients with HME, popliteal pseudoaneurysms is more frequently diagnosed in the context of a non–traumatic event (65%) than after trauma (35%), and symptoms consist of pain and swelling on the popliteal fossa with preserved distal blood flow [64]. Nevertheless, the presence of ecchymosis or pulsatile mass in the popliteal region is not frequently observed but is highly suggestive of a popliteal pseudoaneurysm [64]. Duplex ultrasound is the gold of standard for diagnosis of pseudoaneurysms because it can clearly depict the vascular lesion and its relationship with the osteochondroma [58]. However, duplex ultrasound has some limitations in patients with obesity or huge hematomas due to more thickness between the vascular structures and the transductor. In these cases, some authors have argued in favour of angiography to depict the anatomic details before surgery. [51].

With the reported case, as underlined by other Authors, we strongly support the use of doppler ultrasound because it could be used as a functional examination showing the blood flow asset during complete extension of the knee [1,3,29,56,61,62,65]. Given the benign nature of osteochondromas, surgical resection is considered the gold standard of treatment, however we suggest a combined vascular-orthopedic approach from the onset of symptoms and diagnosis [66]. When surgery is planned, a complete excision of the cartilaginous cap is critical to prevent recurrence.

## 6. Treatment

The main consideration about management is how to approach the vascular complication considering that treatment of osteochondroma is mainly simple surgical resection at the base of the tumor with meticulous dissection of the neurovascular structures. Most cases were managed by osteochondroma resection with or without open vascular reconstruction. Open surgical options include bypass and interposition grafting, direct repair, patch angioplasty, and aneurysm resection with direct end-to-end anastomosis. Arterial reconstruction should be considered by either closing the arterial defect in cases of pseudoaneurysm formation or using a saphenous vein graft in cases of arterial thrombosis and extensive arterial damage. From the literature review, pseudoaneurysm resection associated with venous bypass or end-to-end anastomosis represent the most frequent surgical approach, because of the high risk of recurrent pseudoaneurysm or stenosis after direct closure. Vasseur et al. reviewed 97 patients with osteochondroma associated with vascular complications, surgically repaired [19]. Among them, direct closure was performed in 25 cases with a higher risk of recurrent pseudoaneurysm if stress is applied to the suture line (as the patient ages). Moreover, revascularization is considered the gold standard surgical option for preventing tissue damage or limb loss from embolic events in the setting of acute limb ischemia [3]. Most of the surgeons suggest a complete removal of the damaged wall and the use of patch or saphenous vein for defect repair, to reduce the rate of late complications.

Endovascular management is infrequently described. In two hybrid procedures, one patient underwent thrombectomy for embolus in the tibioperoneal trunk with subsequent vein patching of the popliteal pseudoaneurysm, and another had coil embolization of the pseudoaneurysm followed by direct repair of the artery one week later [62,63]. Sometimes, treatment of vascular complication consists of emergency surgery. Banno et al. reported a pseudoaneurysm of the popliteal artery in a 16-year-old boy with HMO [55]. Duarte at al. recently reported a 15-year-old patient with a history of current massages as part of his gym routine, who arrived at the emergency department with four days of pain, and ecchymosis in the right popliteal region: duplex ultrasonography and arteriography confirming the diagnosis of popliteal pseudoaneurysm [58]. Nasr at al. described two cases of vascular complications that occurred in a solitary form and the other with HMO, for a lower limb ischemia and pseudoaneurysm of the left superficial femoral artery [56]. Chen at al. reported a large literature review on 130 vascular complications from osteochondroma in all sites. Vascular complications were due to osteochondromas of the femur (66.2%), tibia (15.4%), fibula (7.7%), humerus (5.4%), clavicle (1.5%), ribs (1.5%), pubic ramus (0.8%), scapula (0.8%) and cervical vertebra (0.8%). The popliteal artery was involved in 66% of cases, and popliteal pseudoaneurysms were the most prevalent vascular complication (49%) [30]. In the same paper the case of a 38-year-old man was reported, presenting with left calf pain and swelling due to a popliteal pseudoaneurysm and incidental peroneal vein thrombosis secondary to a fractured femoral sessile osteochondroma, treated with resection of the osteochondroma, excision of the aneurysm, and primary end-to-end anastomosis of the artery [30].

Local recurrence of osteochondroma after resection is low at 1.8% but may be higher in skeletally immature patients [26,30]. Two cases of delayed pseudoaneurysm formation after incomplete osteochondroma resection have been reported. In one case a partially resected osteochondroma resulted in pseudoaneurysm formation five years later because of ongoing arterial trauma from the residual tumor’s irregular edges [66]. Scotti et al. reported brachial pseudoaneurysm presenting with rupture secondary to a partially resected osteochondroma three years earlier [67]. In our case, combined vascular-orthopedic approach was initiated with intra-arterial locoregional thrombolytic therapy in place for four days and for the surgical tangential resection of the tibial osteochondroma.

## 7. Conclusions

Osteochondromas are the most common bone tumors. Most are asymptomatic and discovered incidentally. This frequent benign tumor may occasionally cause severe arterial complications requiring surgical urgent procedures. CT scan and MRI with contrast medium are adequate for diagnosis, even if simple functional doppler ultrasound is able to show clearly the correlation between vascular compression and ostheocondroma. A combined vascular-orthopedic approach is mandatory from diagnosis to surgical management, and it is crucial for limb salvage procedure in cases of vascular involvement.

## Figures and Tables

**Figure 1 diagnostics-12-01191-f001:**
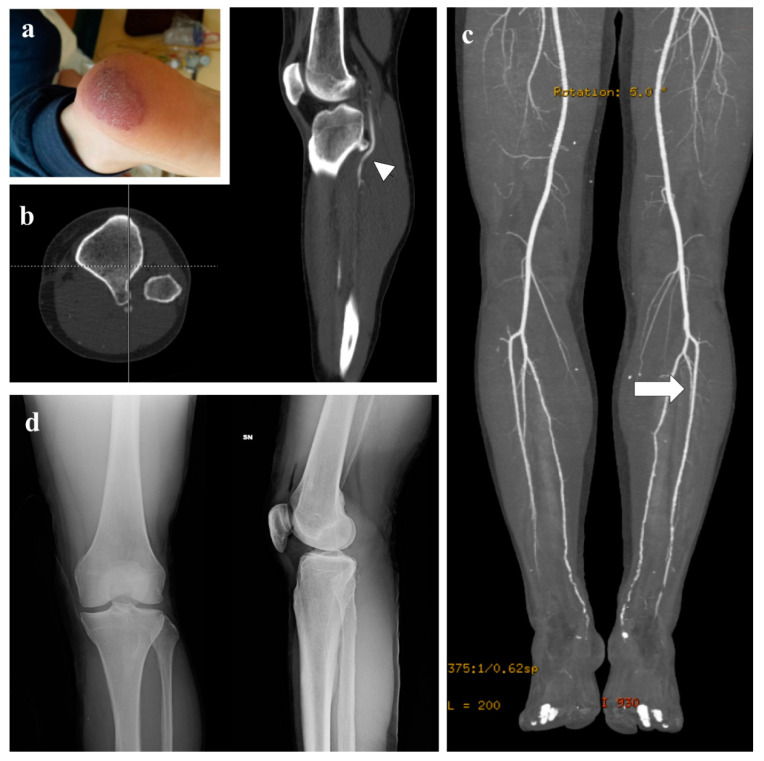
(**a**) Ecchymosis at left heel. (**b**) Typical imaging appearance at axial and sagittal CT scan, with pedunculated osteochondroma with posterior extension and compression of arterial bundle (arrow head). (**c**) CT Angiogram show thrombosis of the subarticular popliteal artery (white arrow). (**d**) antero-posterior and lateral radiograph appear less representative for clear diagnosis of the bony lesion.

**Figure 2 diagnostics-12-01191-f002:**
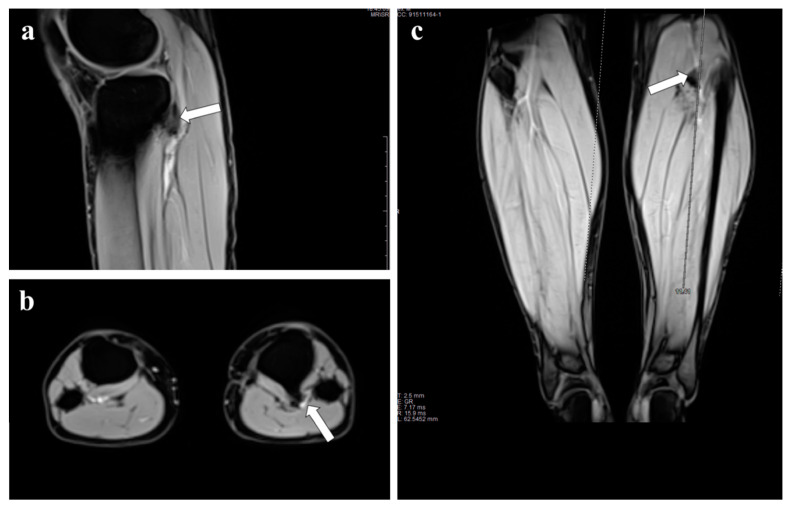
Magnetic resonance (**a**) sagittal, (**b**) axial and (**c**) coronal STIR images show the marrow and cortical continuity of the pedunculated osteochondroma and underlying tibia, and with a relatively thin cartilage cup (white arrows).

**Figure 3 diagnostics-12-01191-f003:**
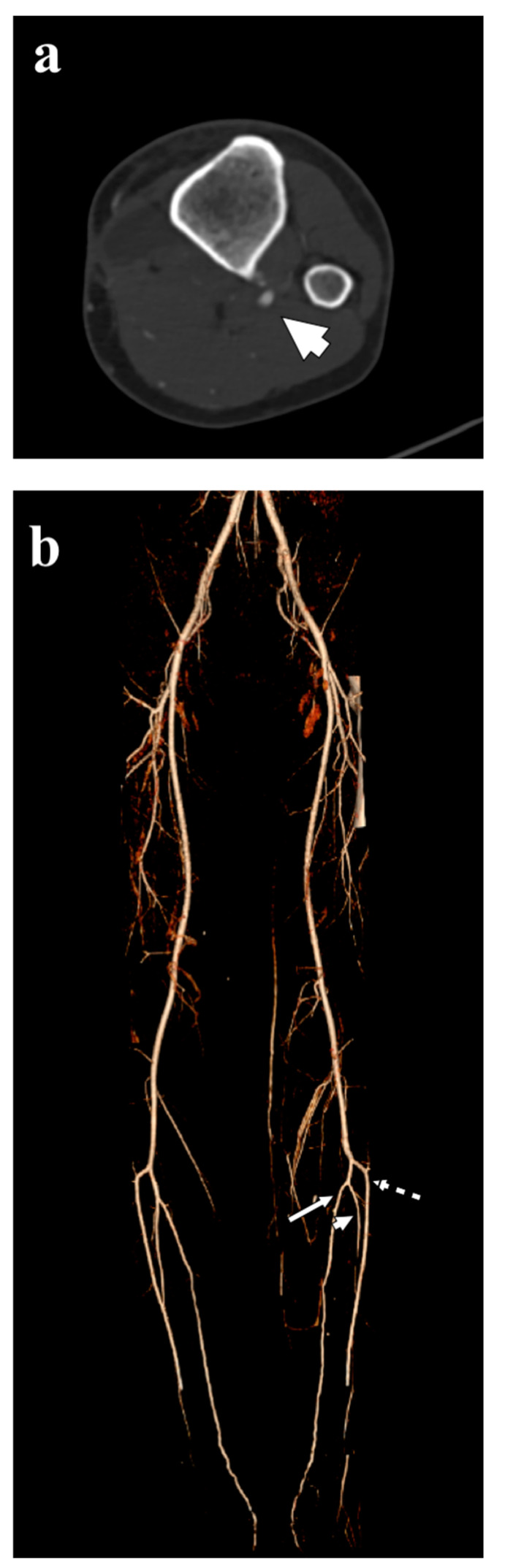
Postoperative examination. (**a**) Axial CT scan with contrast show the complete removal of the tumor and the vascular bundle (arrowhead). (**b**) Three-dimensional volume rendered CT angiography image shows complete revascularization of the anterior tibial artery (dashed arrow), peroneal artery (arrowhead) and posterior tibial artery (flat arrow).

**Table 1 diagnostics-12-01191-t001:** Summary of the published reports on osteochondroma presenting with lower extremity vascular symptoms and complications.

Authors	Pts	Symptoms	Bone Site	Treatment of Vascular Lesion	Age (Years)	Exostosis
Masson et al. [9]	1	S + P	Femur	Popliteal artery ligature + VB	9	Multiple
Refs. [10,11]	2	S (1), P (1) AI (1)	Femur	PAR + VB	Mean 31.5 (15–48)	Solitary
Refs. [12,13,14]	4	S (2), P (2),AI (2)	Femur	n.a.	Mean 22 (13–39)	Solitary
Refs. [15,16,17,18,19,20]	7	S (5), P (4)	Femur	Direct repair	Mean 19.7 (15–20)	Multiple
Shah et al. [21]	1	AI	Femur	PAR + direct repair	16	Solitary
Refs. [22,23,24]	3	S (2), P (2), AI (1)	Tibia	Flattening + VB	Mean 22 (20–23)	Solitary
Refs. [25,26]	2	S (1) P (2)	Femur	PAR + EtoEA	Mean 18.5 (16–18)	Multiple
Refs. [17,19,27,28,29,30]	6	S (6), P (4)	Femur	PAR + EtoEA	Mean 34.3 (14–51)	Solitary
Refs. [31,32,33,34]	4	S (2), P (4)	Femur	Ligature + direct repair	Mean 15.3 (9–16)	Solitary
Refs. [35,36,37]	3	S (3), P (1) AI (1)	Femur	Flattening + VB	Mean 14.3 (13–16)	Solitary
Refs. [38,39,40,41,42]	5	S (3), P (3), Pulsatile mass (1)	Femur (4) Fibula (1)	Vein patch	Mean 18.8 (13–33)	Solitary
Refs. [19,38,43]	4	S (3), P (2), AI (1)	Femur	Ligature + VB	Mean 22.3 (14–37)	Solitary
Hasselgren et al. [44]	1	S + P	Femur	Ligature + prosthetic bypass	45	Multiple
Smits et al. [13]	1	Calf swelling	Tibia	n.a.	28	Solitary
Wiater et al. [45]	1	P	Femur	VB	17	Multiple
Cardon et al. [46]	1	P	Femur	Resection + EtoEA	12	Solitary
Toth et al. [47]	1	S + P	Femur	Flattening + prosthetic bypass	17	Multiple
Oxenius et al. [48]	1	S + P	Femur	Pericardial patch	13	Solitary
Refs. [49,50,51,52,53,54,55]	8	S (5), P (4), C (2), AI (1)	Femur (8)	Flattening + VB	Mean 17.5 (12–21)	Multiple
Nasr et al. [56]	1	AI	Femur	Vein patch + EtoEA	17	Multiple
	1	S + P	Femur	Flattening + EtoEA	17	Multiple
Refs. [57,58]	2	S (1), P (2), C (1)	Tibia	Direct repair	Mean 14.5 (14–15)	Multiple
Present case	1	AI	Tibia	Thrombolytic Therapy	34	Solitary

VB: venous bypass; PAR: Pseudoaneurysm resection; EtoEA: end-to-end anastomosis; n.a.: not available; S: Swelling; P: pain; AI: Acute ischemia; C: Claudication.

## Data Availability

Manuscript data are embedded in the text and fully available on specific request.

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
