# Peer review of "Vascular Complications Caused by Tibial Osteochondroma: Focus on the Literature and Presentation of a Popliteal Artery Thrombosis with Acute Lower Limb Ischemia"

_diagnostics, 2022, doi:10.3390/diagnostics12051191_

Round 1
Reviewer 1 Report
Very interesting case-based review:
I suggest some minor improvements:
- Figure 2 please include in legends the sequences shown. Please add one or more arrows, particularly to show the cartilagineous cap.
- Figure 1 please remove the thin arrows yellow and green in the upper right corner of Panel C. In Panel b this is not an arrowhead, please change it or change the legend.
- The paragraph 0. How to Use This Template must be removed.
- The table with 60 references reported is huge and not so useful for the readers. I suggest to reduce this. May you put together cases/case reports with same characteristics so reducing the number of lines in the table?
Author Response
Thank you Reviewer 1 for your comments. The requested corrections have been highlighted in red.1) Figure 2 please include in legends the sequences shown. Please add one or more arrows, particularly to show the cartilagineous cap.
R1) Thank you Reviewer 1 for your comments. We added the informations in the figure legend, adding the requested arrows
Figure 2. Magnetic resonance a) sagittal, b) axial and c) coronal STIR images show the marrow and cortical continuity of the peduncolated osteochondroma and underlying tibia, and with a relatively thin cartilage cup (white arrows).
2) Figure 1 please remove the thin arrows yellow and green in the upper right corner of Panel C. In Panel b this is not an arrowhead, please change it or change the legend.
R2) Thank you Reviewer 1 for your comments. We changed the figure as requested
3) The paragraph 0. How to Use This Template must be removed.
R3) Thank you Reviewer 1. We are sorry for the mistake. The paragraph has been removed
4) The table with 60 references reported is huge and not so useful for the readers. I suggest to reduce this. May you put together cases/case reports with same characteristics so reducing the number of lines in the table?
R4) Thank you Reviewer 1. The table has been reduced as suggested.
Reviewer 2 Report
The authors reported a case of osteochondroma with vascular complications and with review of literature.
- Though interesting, the manuscript is like a case report with literature review, not a review article.
- Though rare, the manuscript put few information on the diagnosis and treatment in this field as a review article.
Author Response
Reviewer 2.
1) Though interesting, the manuscript is like a case report with literature review, not a review article. Though rare, the manuscript put few information on the diagnosis and treatment in this field as a review article
R1)
Thank you Reviewer 2 for your comment. As per editorial request, we performed a wide literature review on the topic including a report as an explanatory case. However, we agree with your comment and we added new information on the diagnosis and treatment in these cases.
Pag. 7, line 150
- Diagnosis
Pedunculated osteochondromas around the knee have a typical appearance, ex-tending away from the adjacent joint, described as stalactites and stalagmites. In any bone that develops from enchondral ossification, osteochondromas have a marrow, cortical and periosteum continuity with the underlying native bone. This direct continuity clearly appreciable in CT scan is a pathognomonic finding [60]. The cartilaginous cup is the most important area that should be investigated. On MRI, the cartilaginous cap is typically hypointense on T1-weighted images and variable on T2 images (hypointense or hy-perintense if densely or poorly mineralized, respectively) [60]. Angiography and other vascular studies should be considered in presence of vascular compression or for surgical planning [7,11,31,35]. Sometimes these exams are helpful to characterize neovascularity, which may signal malignant transformation [7,35]. It is important to include bone tumors in differential diagnosis in patients with vascular symptoms…
Pag. 8 line 186
… With the reported case, as underlined by other Authors, we strongly support the use of doppler ultrasound because it could be used as functional examination showing the blood flow asset during complete extension of the knee [3,56, 62, 64-65, 68-69]. Given the benign nature of osteochondromas,surgical resection is considered the gold standard of treatment, however we suggest a combined vascular-orthopedic approach from the onset of symptoms and diagnosis [70]. When surgery is planned, a complete excision of the cartilaginous cap is critical to prevent recurrence.
Pag. 8 line 202
… From the literature review, pseudoaneurysm resection associated with venous bypass or end-to-end anastomosis represent the most frequent surgical approach, because of the high risk of recurrrent pseudoaneurysm or stenosis after direct closure. Vasseur et al. reviewed 97 patients with osteochondroma associated with vascular complications, sur-gically repaired [35]. Among them, direct closure was performed in 25 cases with a higher risk of recurrent pseudoaneurysm if stress is applied to the suture line (as the patient ages). Moreover,revascularization is considered the gold standard surgical option for preventing tissue damage
or limb loss from embolic events in the setting of acute limb ischemia [3]. Most of the surgeons suggest a complete removal of the damaged wall and the use of patch or saphenous vein for defect repair, to reduce the rate of late complications.
Round 2
Reviewer 2 Report
The authors modified the manuscript well according to reviewers' comments. It is suitable for published in the journal.